# Factors Associated with Household Food Security in Zambia

**William Nkomoki [1], Miroslava Bavorová [2] and Jan Banout [1,*]**

[1] Department of Sustainable Technology, Faculty of Tropical AgriSciences, Czech University of Life Sciences Prague, Kamýcká 129, 165 00 Prague 6-Suchdol, Czech Republic; nkomoki@ftz.czu.cz

[2] Department of Economics and Development, Faculty of Tropical AgriSciences, Czech University of Life Sciences Prague, Kamýcká 129, 165 00 Prague 6-Suchdol, Czech Republic; bavorova@ftz.czu.cz

\* Correspondence: banout@ftz.czu.cz; Tel.: +420-224-384-186; Fax: +420-23438-1829

**Abstract:** Food security is a global challenge and threatens mainly smallholder farmers in developing countries. The main aim of this paper is to determine factors that are associated with food security in Zambia. This study utilizes the household questionnaire survey dataset of 400 smallholder farmers in four districts conducted in southern Zambia in 2016. To measure food security, the study employs two food security indicators, namely the food consumption score (FCS) and the household hunger scale (HHS). Two ordered probit models are estimated with the dependent variables FCS and HHS. Both the FCS and HHS models' findings reveal that higher education levels of household head, increasing livestock income, secure land tenure, increasing land size, and group membership increase the probability of household food and nutrition security. The results imply that policies supporting livestock development programs such as training of farmers in animal husbandry, as well as policies increasing land tenure security and empowerment of farmers groups, have the potential to enhance household food and nutrition security.

**Keywords:** determinants; food security; hunger; policies; smallholder

---

## 1. Introduction

Food insecurity and undernourishment are on the rise worldwide, from an estimated 777 million people in 2015 to 815 million people in 2016 [1]. This increase is a global concern in achieving the second sustainable development goal, which calls for a commitment to end hunger, reduce food insecurity, and improve nutrition by 2030 [1]. The majority of food-insecure populations reside in Africa, which is home to the largest number of the poorest and most poverty-stricken countries in the world [2]. Zambia is not spared, as the global hunger index report (GHI) ranks Zambia under the category of alarming levels of hunger [3]. In Zambia, the predominant livelihood activity is smallholder farming, mainly cultivating maize and livestock raring. The country's main labor force is agriculture, which employs 72% of the national population. Furthermore, the smallholder farmers are adversely affected by food insecurity.

Previous studies that dealt with determinants of food security considered the following: (i) household head characteristics comprising gender, age, education, farming experience, and marital status; (ii) household characteristics constituting of incomes, livestock ownership, and employment status; (iii) farm characteristics including land size and land ownership; and (iv) institutional characteristics, including access to credit, farmers groups, and extension services, which are then detailed in the conceptual link on determinants of food security. The current study builds on and extends the study by Nkomoki et al. [4] that determine factors that influences the adoption probability of sustainable agriculture practices (SAPs), considering the effect of land tenure, and test the association

between SAPs use and food security in Zambia. The findings of the study reveal that land tenure contributed to adoption of SAPs and the chi-square tests indicated that adoption of SAPs contributed to food security status. However, we acknowledge that results in distribution of food security scores can not only be attributed to land tenure alone as other factors can play an important role. Therefore, to further understand households' food security drivers, we follow up with the ordered probit regression model analysis.

The literature investigating the determinants of food security in Zambia is limited. To the best of the researchers' knowledge, there is scarcity in the literature on this topic; yet, it is a critical subject in Zambia. Therefore, this study aims to consolidate past studies to add an incremental contribution with focus on examining the effect of chosen factors as influencers on food security. The significance of the study is to provide information to policy-makers, so that they can gain an understanding of a range of factors that potentially promote food security.

The structure of the paper is as follows: the next section covers a review of the literature relating to conceptual links on factors affecting food security. In the third section, the study area, data collection, and methods of data analysis are described, followed by the results and discussion. The paper ends with conclusions.

## 2. Conceptual Link on Determinants of Food Security

Following prior literature, several factors are associated and considered as determinants of food security. Holden and Ghebru [5] contended that, for smallholder farmers, the ultimate goal is to achieve food security.

There are a number of studies that investigated the effect of gender of household head on food security. Mallick and Rafi [6] examined the food security status of male- and female-headed households in Bangladesh. Their results revealed that gender of household head had no effect on household security and this was attributed to no cultural and social restriction for women's participation in labor force. A study by Kassie et al. [7] assessed how gender of household heads was associated with food security in Kenya. They documented that female-headed households were more vulnerable to food insecurity than male-headed households. Similarly, Tibesigwa and Visser [8] evaluated the impacts of gender inequality among smallholder households in South Africa on food security. Their results revealed that male-headed households were more food-secure when compared to their female counterparts. Further, they indicated that a wider gap in food security was observed in rural areas in contrast to the households in urban areas [8].

De Cock et al. [9] investigated the household food security situation in rural South Africa. The multivariate analyses indicated that education of household head positively contributed to food security. Maitra and Rao [10] examined the factors affecting household food security in Kolkata, India. The findings of the ordered probit model revealed that a household head with higher education level increased the chance of household being food-secure. Using the logistic regression model, Zhou et al. [11] explored the factors that influence food security in rural Pakistan. The results demonstrated that education of the household level played an important contribution toward households being food-secure.

A study by De Cock et al. [9] investigated the determinants of food security in rural South Africa, and the multivariate regression analyses found that household size was a major determinant of household food security, and a smaller household size was less likely to be food-insecure. A study by Kabunga et al. [12] used the Household Food Insecurity Access Scale to measure household food security, and found that larger household sizes are associated with higher food insecurity in Kenya. In contrast, the findings of the study by Maitra and Rao [10] in India indicated that a larger household size had less likelihood to be found in a food-insecure category. The contention is that, with a larger household, the number of bread-winners that the household may depend on for household provision is higher.

In many developing countries, non-farm income is viewed as an opportunity to broaden income base and contribute to food security. However, Owusu et al. [13] indicated the limited number of studies linking the relationship between non-farm income with food security. Babatunde and Qaim [14] focused on the effects of off-farm incomes on food and nutritional security in Nigeria. The results demonstrated that off-farm income positively affected food security and nutrition. Owusu et al. [13] studied the impact of off-farm works on household food security in northern Ghana and found that it positively contributed to household food security. Also, in northern Ghana, Zereyesus et al. [15] evaluated the influence of participating in non-farm income activities on food poverty. They found that involvement in non-agriculture activities improved households' food consumption.

Generoso [16] studied the impacts of remittances on food security in rural Mali. Results of the logistic regression model indicted that households with remittances had better food security status than those without remittances in the Saharan zone, but contended that the benefit to solve food insecurity was temporary. However, in the same study, no statistical association and contribution was observed for remittances on food security in the Sahelian zone, Mali. A study by Fransen and Mazzucato [17] focused on remittances and household wealth for post-conflict households in Burundi. The findings revealed that, for households in the poor wealth category, the remittance receiving household finances increased and the food security status improved. However, when compared to wealthier households, receiving of remittances did not affect the household food security. A systematic review by Thow et al. [18] focused on the impacts of remittances on diets and nutrition, and the studies revealed that households with remittances had better food consumption, minimized vulnerability, and better food security than the households that did not have remittances. Using the ordered logistic regression, Atuoye et al. [19] investigated the impacts of remittances on household food security among rural and urban households in Ghana. The findings demonstrated that rural and urban households that received remittance were more likely to be in the severe food insecurity category than urban households without remittances. Bhalla et al. [20] studied the impacts of cash transfers on household food security in Zimbabwe and revealed that that cash transfer is a major determinant of household food security and diet diversity. The results further demonstrated an improvement in food security for households that are recipients of cash transfers.

Livestock incomes and ownership are viewed as a potential approach to help minimize food security. Dumas et al. [21] investigated the impact of livestock ownership on food consumption in eastern Zambia. The results did not show association between livestock ownership and dietary diversity among the children in Zambia. Using the food security index, Demeke et al. [22] examined the determinants of household food security in rural Ethiopia and found that households with more livestock ownership were less likely to be food-insecure. Mango et al. [23] applied linear regression to evaluate the factors that affect food security among smallholder farmers in Zimbabwe and found that livestock ownership and income contributed to better household food security. Rawlins et al. [24] indicated the importance of livestock in improving nutritional status of rural households in Rwanda. In a study in sub-Saharan Africa, Hetherington et al. [25] demonstrated that livestock ownership is associated with increased food consumption.

Agricultural land ownership is another factor identified in previous studies as associated with food security. Robertson and Pinstrup-Andersen [26], for example, argued that food security is threatened as the majority of smallholder farmers lack formal users' rights to agricultural land in developing countries. Similarly, Headey and Jayne [27] noted that issues of land constraints are of relevance in Africa, and the land tenure systems are part of that concern to ensure food security [28]. In addition, Holden and Ghebru [5] recognized that enhanced agriculture and productivity eventually lead to improved food security. A study that Chirwa [29] conducted in Malawi focused on land tenure systems and food production, and the results indicated that households that benefited from land reform programs to strengthen their land tenure security from customary land reported high maize production and an increase in food security when compared to non-beneficiaries in customary land. Simbizi et al. [30] e investigated the role that tenure security plays in rural, poor sub-Saharan Africa, and their findings

indicated that land security is a major determinant of food security. Mwesigye et al. [31] indicated that private ownership yielded higher crop outputs when compared to customary land tenure systems in Uganda. A private owner was considered to have secure land rights, compared to the customary land tenure with limited land use rights. Michler and Shively [32] studied the relationship between land tenure and efficiency in farm productivity in the Philippines, and their findings indicated that land tenure contributed to farm productivity. Moreover, research by Mendola and Simtowe [33] in Malawi indicated that access to a secure productive resource such as land enhances food security. A study by Santos et al. [34] analyzed the land allocation and registration program in India's West Bengal to evaluate whether government-allocated land contributed to food security. Their findings revealed that no statistical association was observed to impact food security from government land. For Zambia, less research was carried out to explore the land tenure systems [35]. The study by Smith [36] in Zambia demonstrated that formal land titles enhanced investment and were more profitable, with higher output on agricultural productivity. Merten and Haller [37] studied the role of property rights on child growth and the food security of households in customary land tenure in Zambia, and they found that insecure property rights in the form of type of land tenure affected the food consumption pattern of the households. Sitko et al. [38] analyzed the effects of land titling among smallholder farmers with the objective of determining whether it enhanced growth in agriculture. Their results did not demonstrate any statistical differences between title and non-title holders.

With regard to the influence of farm size on welfare outcomes, Khonje et al. [39] studied the effect of adopting improved maize varieties on welfare outcome indicators, namely food security, poverty, crop income, and consumption expenditure, in the eastern province of Zambia. The findings on farm size and poverty revealed an inverse relationship. Households with a smaller farm size of 0.1–3.5 hectares showed higher poverty levels in 54% cases as compared to households with more than 3.5 hectares, where the poverty levels were in 33% cases. Frelat et al. [40] showed that farm size is a determinant of food security in sub-Saharan Africa. The relationship showed that, as farm size increases, the probability of a household being food-secure also increases. A study by Koirala et al. [41] investigated the role of land ownership on productivity among rice farmers in the Philippines and found that a 1% increase in farm size increased the rice yield by 0.40%. Paul and Wa Githinji [42] examined the relationship between farm size and productivity in Ethiopia. The results demonstrated an inverse negative association between farm size and per hectare yield.

Agricultural group membership is viewed as a vital institution and pathway for smallholder farmers to participate in markets, raise incomes, and eventually reduce poverty. The group membership can for example provide networking and connections which may empower individuals or groups with various business ventures to enhance income generation, and nutritional programs to address issues of food insecurity. Fischer and Qaim [43] investigated the role of agricultural cooperatives among smallholder banana farmers in Kenya. They found that members of farmer groups marketed their produce collectively, yielded a higher price, and had higher income than non-members who marketed individually. Verhofstadt and Maetens [44] analyzed the benefits of farmer membership in a cooperative on poverty in Rwanda. The results of the propensity matching score demonstrated that farmers that belonged to cooperatives had better income and reduced levels of household poverty. Furthermore, Verhofstadt and Maetens [44] argued in support of agricultural group as they are related to collective participation and more inclusive than other innovations that focus on individuals. Abate et al. [45] focused on the impacts of agricultural cooperatives on enhancing the efficiency of smallholder farmers in Ethiopia. The findings showed that agricultural cooperatives contributed to higher farm productivity. Ma and Abdulai [46] analyzed the effect of cooperative membership on household welfare of apple farmers in China and found members of farmers groups to have better farm yields and household income. Mojo et al. [47] evaluated the determinants and economic benefits of membership in coffee cooperatives in rural Ethiopia. The results indicated that membership to a cooperative positively contributed to household incomes.

Concerning access to credit, Aidoo et al. [48] investigated the determinants of household food security in rural Ghana. The results of the logistic regression model analysis revealed that access to credit had a positive influence on a household's food security. In Nigeria, Awotide et al. [49] examined the effect of access to credit on agricultural productivity among smallholder farmers and demonstrated that households with access to credit had higher cassava productivity. In support of alleviating the constraints related to credit access for smallholder farmers, Tirivayi et al. [50] argued for agricultural interventions of establishing microcredit and microfinance institutions in rural areas.

Given the lack of consensus on indicators to measure food security, Carletto et al. [51] suggested that a useful approach is to assess the food security situation of each dimension and specify the level (national, regional, or household). In addition, the research of Headey and Ecker [52], in agreement, revealed that, in measuring food security, a criterion to gauge the indicators is based on the demand of decision-makers for a wide range of information. Vaitla et al. [53] argued that, rather than focusing on one indicator, the best way to capture the food security measurement is to see the complementarity. For this reason, considering one indicator alone cannot necessarily reflect the food situation. Therefore, two indicators were incorporated to measure food security, namely the food consumption score (FCS) and the household hunger scale (HHS).

## 3. Data and Methodology

### 3.1. Study Area

The study area comprised four districts, namely Choma, Mazabuka, Kalomo, and Chikankata, in the southern province of Zambia (Figure 1). In Zambia, agriculture employs 72% of the country's labor force, with more than 60% residing in rural areas [54]. The predominant livelihood activity is smallholder farming, mainly cultivating maize and livestock raring. The study area is classified under moderate rainfall patterns characterized with approximately 800–1000 mm of annual precipitation. The soils in the region are characterized as sand loamy and clay loams. The farming system integrates crop production and livestock rearing as a mixed type of farming. Smallholder crop production includes cereals, tubers, and legumes. Cash crops such as sunflower, cotton, tobacco, and soya beans are also cultivated. They also rear livestock, mainly cattle, goats, and poultry. The communal lands are open for livestock grazing usually after crop harvests, while, at the same time, the land tenure rights are respected. Regarding cultural characteristics, the study area is home to the Tonga people who are the main ethnic group. In Tonga culture, the number of cattle owned defines the social status. Households keep and sell goats, pigs, and poultry to be able to pay for immediate needs such as health bills and education. In the survey area, the farmers learned the different aspects of agriculture mainly by sharing their knowledge through networking in farming groups and/or information dissemination by extension services. Agriculture extension support is further coordinated by the Ministry of Agriculture and cooperatives through extension workers. It provides agriculture-related information on the television and radio, and organizes agriculture shows at the district, provincial, and national level. Agriculture extension is important as it helps farmers decide on whether to choose new technologies and increase production.

In Zambia, agricultural land ownership is categorized into two regimes, namely (i) the customary land tenure that accounts for 60%, and (ii) the statutory land tenure that accounts for 40%. In customary tenure, land is controlled by the traditional leaders in the communities. This land is informally recognized, and it lacks tenure security that results in it having limited land users' rights, which increases the chances of farmer eviction from the land. In contrast to customary tenure, a statutory land tenure is issued with land titles that indicates exclusive ownership, full land rights, and protection from eviction [55]. Zambia has a total land mass of 752,621 km$^2$ [56]. Despite the abundance of land, the possibility of agricultural growth is increasingly challenging due to smallholder farmers' limitations to land access [57]. The policies on land in Zambia remained stagnant for decades, as the policy-makers often do not consider the smallholder farmers' land constraints [58]. The national development plan

report for 2017–2021 indicated that there is low access to land in Zambia, despite it being a vital resource for investment, the creation of wealth, and ultimately contributing to poverty reduction [59]. To reduce the challenges that smallholder farmers face in accessing land, strategies for assessments of land distribution and governance play an important role [60].

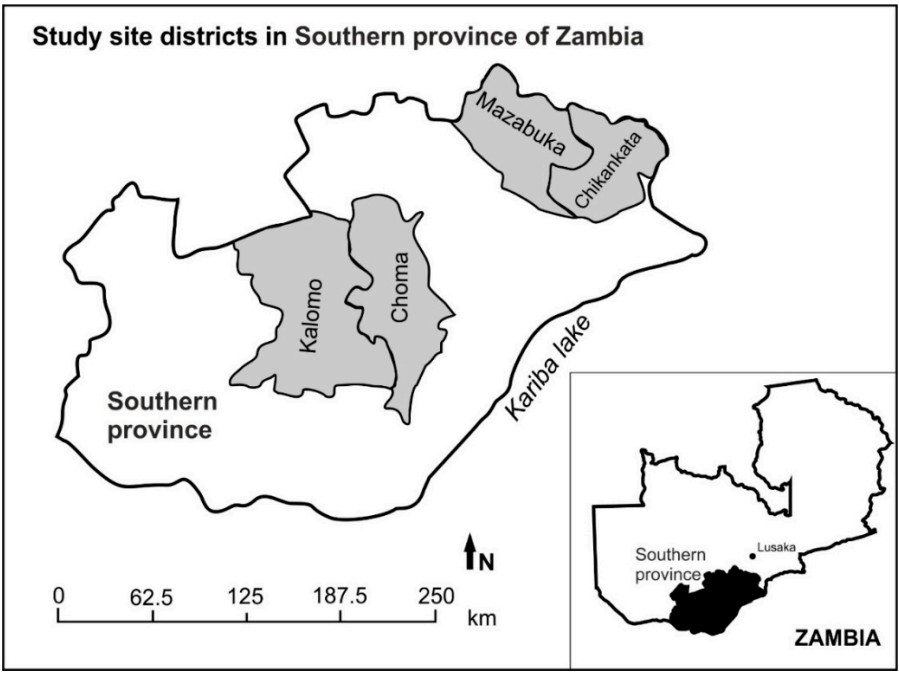

**Figure 1.** Study site.

## 3.2. Data Collection and Sample

The data were derived from a household survey conducted in southern Zambia in 2016. The study was based on face-to-face interviews with smallholder farmers using a structured questionnaire. The data were recorded on the paper questionnaires (pen-and-paper personal interview) and later coded on Microsoft Excel spreadsheets. The household heads were targeted in the interviews and, in cases where the household head was absent, the next household head (for example, the wife) was considered. The provinces and districts were purposively selected. The region was selected because, even though the area is regarded as the food basket of the country, the population still faces food insecurity, which makes it suitable for the study [61]. A total of six villages were sampled per district. Three villages were selected per tenure system in each district, using a systematic approach that was guided by the following key features: (i) villages in different locations, and (ii) villages with comparable tenure systems. One hundred small holder farmers per district were considered, to draw a total sample of 400 farm households—200 under statutory and 200 in customary land tenure systems—who were randomly selected. The questions were related to socioeconomic and demographic characteristics, consumption patterns, household hunger, and livelihood activities. The questionnaire was pretested before actual data collection by the authors and local trained enumerators. Apart from the questionnaire, qualitative data were gathered through in-depth interviews conducted with village headmen, officials from the Ministry of Agriculture, and the Ministry of Community and National Development district offices in Zambia. Research consent was obtained from the district commissioner's office.

### 3.3. Research Variables

#### 3.3.1. Food Security Indicators

The FCS was developed by the World Food Program as a frequency-weighted dietary diversity score [62]. Different studies applied the FCS indicator in Tanzania [63], Rwanda [64], and Kenya [65].
The FCS is calculated as follows [66]:

$$FCS = a\_1\, b\_1 + a\_2\, b\_2 + \ldots a\_8\, b\_8, \tag{1}$$

where $a$ = frequency (one-week recall period), 1–8 = food group, and $b$ = weight (meat, milk, and fish = 4; pulses = 3; staples = 2; vegetables and fruits = 1; and oil and sugar = 0.5).

The threshold for the FCS classifies households into one of the following categories: poor (<21.5), borderline (21.5–35), and acceptable (>35).

The HHS was developed by the Food and Nutrition Technical Assistance. It is a cross-culturally validated food security indicator that captures elements of cultural experiences and severe food insecurity, and it was applied across studies conducted in Kenya, Zimbabwe, South Africa, Mozambique, Malawi, and the Gaza Strip [66,67]. A four-week recall period is set as standard in data collection. The HHS questionnaire consists of the following three questions: (i) Was there ever no food at all in your household because there were no resources to get more? (ii) Did you or any household member go to sleep at night hungry because there was not enough food? (iii) Did you or any household member go a whole day and night without eating anything because there was not enough food? The responses to the questions were classified as follows: rare = 0 (twice a month), sometimes = 1 (three to 10 times), and often = 2 (>10 times). The values were added up for the three questions, and the scores ranged from 0–6. The HHS categories are as follows: little to no hunger (scores 0–1), moderate hunger (scores 2–3), and severe hunger (scores 4–6) [62].

#### 3.3.2. Ordered Probit Model

The ordered probit regression model was used to examine the effect of the chosen factors as influencers on food security.

#### Ordered Probit Model

The dependent variables are categorical and ordinal; therefore, the ordered probit regression model is more suitable for the analysis than multinomial regression or a probit regression model [68].
The ordered probit model regression is calculated with the following equation:

$$y\_\hat{i}* = x_i\beta \; + \; \varepsilon\_i, \tag{2}$$

where $y_i^*$ is an unobserved random variable, $x$ is a vector of socioeconomic variables assuming normal distribution, $\varepsilon_i = N\,(0, 1)$, and $i = 1, 2, \ldots , N$.
$y_i$ is the observable ordinal variable, $y_i = j$ if $\mu_{j-1} < y_i^* \le \mu_j$,,
where $j = 0, 1, \ldots , n$, $\mu_{-1} = -\infty$, and $\mu_n = +\infty$.
The probability is calculated with the following interval decision rule:

$$Prob\,[y_i \; = j] = \; \Phi\big(\mu_j - \; x_i\beta\big) - \Phi\big(\mu_{j-1} - \; x_i\beta\big), \tag{3}$$

where $\Phi$ denotes the cumulative distribution function, and $j$ represents the categories of dependent variables.

Dependent Variables

The dependent variables were the FCS and the HHS food security indicators. The FCS indicator is ordered into three categories, namely poor, borderline, and acceptable. The HHS is also classified into three categories: severe hunger, moderate hunger, and little to no hunger.

Explanatory Variables

The selection of explanatory variables was based on findings of previous research. The variables were classified into four groups: (i) household head characteristics, (ii) household characteristics, (iii) farm characteristics, and (iv) institutional characteristics. The household head variables included gender, age, education level, marital status, and farming experience; household characteristics included household size, self-employment, remittances, and off-farm and livestock income; farm characteristics included land ownership and land size; and institutional characteristics included access to credit and membership to farmer groups. The variables were tested for multicollinearity. The variance inflation factor (VIF) values were in the range lower than 10, indicating no multicollinearity problems.

The Statistical Package for Social Sciences IBM (SPSS) and STATA software were used for the data analysis.

## 4. Results

### 4.1. Description of Model Variables

The model variables used in this study are presented in Table 1. The mean FCS value was 27, while the HHS had a mean value of 1.5. In relation to gender of household heads, the majority of the households were led by men, accounting for 63%. The level of education demonstrated that 22.3% of the household heads did not have any form of education, while 37% and 39% were indicated as having primary and secondary education, respectively. The average size of the household was made up of seven members. In this study, the non-farm incomes were divided as self-employment activities that included business activities such as shop-keeping, charcoal sales, and hand crafts, while the off-farm activities included formal and informal non-agricultural wages. In this category, 42.7% of households indicated off-farm income and 26% of the households received remittances. Livestock ownership is an important asset in the southern province. As many as 61.5% of the households indicated livestock ownership, ranging from poultry to pigs, goats, and cattle. Some sold the livestock to boost their subsistence income. The average livestock income was 1087 Zambian kwacha. Agricultural land ownership was categorized as statutory and customary tenure system. The average land size of smallholder farmers was 3.2 hectares; however, the majority were categorized under less than a hectare. Access to credit was constrained, and this could be attributed to poor establishment of financial institutions targeting smallholder farmers. More than 50% of the smallholder farmers were indicated as participating in farmers groups.

**Table 1.** Description of variables.

| Variable | Description | Mean (*n* = 400) |
|---|---|---|
| *Food security indicators* | | |
| Food consumption score (FCS) | Three categories: poor (<21.5), borderline (21.5–35), acceptable (>35) | 26.94 (19.91) |
| Household hunger scale (HHS) | Three categories: little to no hunger (0–1), moderate hunger (2–3), severe hunger (4–6) | 1.52 (0.74) |

**Table 1.** *Cont.*

| Variable | Description | Mean (*n* = 400) |
|---|---|---|
| *Household head characteristics* | | |
| Gender | Sex of household head (male = 1) | 63.0% |
| Age | Number of years for household head | 40.81 (13.30) |
| Education level | 0 = none, 1 = primary, 2 = secondary, 3 = tertiary | 0 = 22.3% |
| Farming experience | Number of years spent in farming | 10.00 (9.79) |
| Marital status | Married = 1 | 85.8% |
| *Household characteristics* | | |
| Household size | Number of members | 6.70 (3.26) |
| Self-employment | Household has business (yes = 1) | 51.5% |
| Remittances | Family received money from relatives (yes = 1) | 26.30% |
| Off farm | Household has salaried or waged incomes (yes = 1) | 42.70% |
| Livestock income | Household has an income from livestock sales (Zambian kwacha) | 1087.34 (2947.59) |
| *Farm characteristics* | | |
| Land ownership | 1 = statutory, 2 = customary | *n* = 400 |
| Land size | Size of agricultural land in hectares | 3.26 (2.82) |
| *Institutional characteristics* | | |
| Access to credit | Household has access to credit (yes = 1) | 16.30% |
| Member of farming group | Household belongs to farming group (yes = 1) | 51.50% |

Note: The mean values are reported with the standard deviation in parentheses. Percentages are reported as indicated; $1 United States dollar (USD) = 10 Zambian kwacha.

## 4.2. Influencers on Food Security

The results of the ordered probit models of the factors affecting food security are presented in Tables 2 and 3.

**Table 2.** Ordered probit regression model (FCS).

| Variables | Coefficient | Food Consumption Score | | |
|---|---|---|---|---|
| | | Poor | Borderline | Acceptable |
| *Household head characteristics* | | | | |
| Gender | 0.098 (0.146) | −0.026 (0.039) | 0.013 (0.020) | 0.026 (0.038) |
| Age | −0.011 (0.007) | 0.003 (0.002) | −0.002 (0.001) | −0.003 (0.002) |
| Education level | 0.472 *** (0.092) | −0.126 *** (0.023) | 0.061 *** (0.016) | 0.126 *** (0.025) |
| Farming experience | −0.012 (0.010) | 0.003 (0.003) | −0.002 (0.001) | −0.003 (0.003) |
| Marital status | −0.288 * (0.161) | 0.077 * (0.043) | −0.037 * (0.023) | −0.077 * (0.042) |
| *Household characteristics* | | | | |
| Household size | 0.084 *** (0.025) | −0.023 *** (0.006) | 0.011 *** (0.004) | 0.022 *** (0.007) |
| Self-employment | 0.021 (0.138) | −0.006 (0.037) | 0.003 (0.018) | 0.006 (0.037) |
| Remittances | −0.256 (0.157) | 0.068 (0.042) | −0.036 (0.024) | −0.064 * (0.038) |
| Off farm | −0.388 *** (0.085) | 0.104 *** (0.021) | −0.050 *** (0.014) | −0.104 *** (0.023) |
| Livestock income | 0.000 *** (0.000) | −0.000 *** (0.000) | 0.000 *** (0.000) | 0.000 *** (0.000) |

**Table 2.** *Cont.*

| Variables | Coefficient | Food Consumption Score | | |
|---|---|---|---|---|
| | | Poor | Borderline | Acceptable |
| *Farm characteristics* | | | | |
| Land ownership | −0.485 *** | 0.129 *** | −0.061 ** | −0.129 *** |
| | (0.137) | (0.035) | (0.020) | (0.037) |
| Land size | 0.091 *** | −0.024 *** | 0.012 *** | 0.024 *** |
| | (0.026) | (0.007) | (0.004) | (0.007) |
| *Institutional characteristics* | | | | |
| Access to credits | 0.128 | −0.034 | 0.016 | 0.036 |
| | (0.185) | (0.049) | (0.021) | (0.053) |
| Farming group member | 0.301 ** | −0.081 ** | 0.039 ** | 0.079 ** |
| | (0.143) | (0.038) | (0.019) | (0.038) |
| Cut1 | 0.431 | | | |
| | (0.363) | | | |
| Cut2 | 1.202 | | | |
| | (0.367) | | | |
| Number of observations | 400 | | | |
| Prob > chi$^2$ | 0.000 | | | |
| Pseudo $R^2$ | 0.264 | | | |

Note: *** $p < 0.01$, ** $p < 0.05$, * $p < 0.1$. The average marginal effects are reported with the standard errors in parentheses.

**Table 3.** Ordered probit regression model (HHS).

| Variables | Coefficient | Household Hunger Scale | | |
|---|---|---|---|---|
| | | Severe Hunger | Moderate Hunger | Little to No Hunger |
| *Household head characteristics* | | | | |
| Gender | 0.009 | 0.001 | 0.002 | −0.002 |
| | (0.147) | (0.014) | (0.034) | (0.036) |
| Age | 0.007 | 0.001 | 0.002 | −0.002 |
| | (0.006) | (0.001) | (0.002) | (0.002) |
| Education level | −0.468 *** | −0.044 *** | −0.120 *** | 0.116 *** |
| | (0.096) | (0.012) | (0.025) | (0.022) |
| Farming experience | −0.003 | −0.000 | −0.001 | 0.001 |
| | (0.010) | (0.001) | (0.002) | (0.002) |
| Marital status | 0.010 | 0.001 | 0.002 | −0.002 |
| | (0.103) | (0.010) | (0.024) | (0.026) |
| *Household characteristics* | | | | |
| Household size | 0.000 | 0.000 | 0.000 | −0.000 |
| | (0.027) | (0.003) | (0.006) | (0.007) |
| Self-employment | −0.101 | −0.010 | −0.024 | 0.025 |
| | (0.142) | (0.013) | (0.033) | (0.035) |
| Remittances | 0.015 | 0.001 | 0.004 | -0.004 |
| | (0.165) | (0.016) | (0.038) | (0.041) |
| Off farm | 0.201 ** | 0.019 ** | 0.047 ** | -0.050 ** |
| | (0.792) | (0.008) | (0.019) | (0.019) |
| Livestock income | −0.000 *** | −0.000 *** | −0.000 *** | 0.000 *** |
| | (0.000) | (0.000) | (0.000) | (0.000) |
| *Farm characteristics* | | | | |
| Land ownership | 0.354 ** | 0.033 ** | 0.086 ** | −0.088 ** |
| | (0.145) | (0.015) | (0.034) | (0.035) |
| Land size | −0.215 *** | −0.020 *** | −0.050 *** | 0.053 *** |
| | (0.044) | (0.005) | (0.011) | (0.010) |

**Table 3.** *Cont.*

| Variables | Coefficient | Household Hunger Scale | | |
|---|---|---|---|---|
| | | Severe Hunger | Moderate Hunger | Little to No Hunger |
| *Institutional characteristics* | | | | |
| Access to credits | −0.077 | −0.007 | −0.018 | 0.019 |
| | (0.219) | (0.019) | (0.050) | (0.054) |
| Farming group member | −0.706 *** | −0.069 *** | −0.159 *** | 0.175 *** |
| | (0.155) | (0.019) | (0.036) | (0.036) |
| Cut1 | −0.6856 | | | |
| | (0.356) | | | |
| Cut 2 | 0.3834 | | | |
| | (0.358) | | | |
| Number of observations | 400 | | | |
| Prob > chi$^2$ | 0.000 | | | |
| Pseudo $R^2$ | 0.270 | | | |

Note: *** $p < 0.01$, ** $p < 0.05$, * $p < 0.1$. The average marginal effects are reported with the standard errors in parentheses.

## 5. Discussion

### 5.1. Household Head Characteristics

In the FCS model, household heads who were more educated were 12.6% less likely to be in the poor FCS category, 6.1% were more likely to be borderline, and 12.5% were more likely to be in the acceptable category of FCS than their less educated counterparts. Regarding the HHS model, our findings indicate that, with an increase in education level, there was a respective 4.4% and 12% lower probability of households being in the severe hunger and moderate hunger categories, while 11.6% had more chance of being in the little to no hunger category. This result is similar to the work of Mason et al. [63], who used the food consumption as an indicator of food security to determine the factors influencing food security in Tanzania. They found that households featuring a household head with a higher education level had better food security status.

### 5.2. Household Characteristics

The households that had off-farm income were 10% more likely to be in the poor FCS category in this study, while 5% were less likely to be in the borderline FCS category and 10% were less likely to be in the acceptable FCS category than those who did not. The HHS indicated that an increase in off-farm activities resulted in the likelihood of a household to be in a severe hunger category being 1.9%, while 4.7% of households were more likely to be in the moderate hunger category, and 5.3% were less likely to be in the little to no hunger category. This finding can be attributed to the fact that households devoted more time to off-farm activities at the expense of farm activities so that they may provide higher food production for their own consumption. With similar results, Mabuza et al. [69] analyzed the impact of income sources on household food insecurity in Swaziland. Their findings reported that on-farm income-dependent households were more food-secure when compared to their counterparts that depended on off-farm income sources. Beyene and Muche [70], in Ethiopia, indicated that off-farm incomes positively contributed to the household food security. The policy aspect would seek how to develop formal employment opportunities that would enhance income levels of the household. The improvement in conditions services would increase the number of people able to acquire food and improve their food security status to substantiate the farm incomes.

According to our results, an increase in livestock incomes was associated with a lower likelihood of being in the poor FCS category, and a higher likelihood of being in the borderline and acceptable FCS categories. Similarly, the HHS demonstrated that an additional increase in livestock income reduced the probability of the household of being in the severe and moderate hunger categories, while it increased the probability of being in the little to no hunger category. The explanation to this result is that ownership of livestock potentially provides meat, milk, and other quality dairy

products, and increases the quantity of nutritional foods for the households. Secondly, livestock sales usually involving live animals enhance income, which may improve the purchasing power of the household. In support of the importance of livestock ownership and incomes to improving food security, Jodlowski et al. [71], who studied the impact of livestock on food security in Zambia, demonstrated that livestock ownership and sales contributed to the household food security through an increase in food consumption expenditure and dietary diversity. Similarly, Kafle et al. [72] studied the role of livestock transfer programs among poor secure households in Zambia. Their result revealed an increase in the financial capacity and household food security status, which was enhanced by training of households in livestock management topics. In contrast to our finding, Silvia et al. [73], who analyzed the determinants of farm household food security in Kenya, Uganda, and Tanzania, found that ownership of livestock did not contribute to the enhancement of household food security.

### 5.3. Farm Characteristics

Regarding land tenure as a determinant of FCS (Table 2), the findings indicated that the households with customary land tenure were 12.9% more likely to be in the poor FCS category, while 6% were less likely to be in the borderline FCS category, and 12.9% were less likely to be in the acceptable FCS category than households with statutory land tenure. Similarly, the effect on land tenure as a determinant of HHS (Table 3) revealed that households with customary land tenure were 3.3% more likely to be in the severe hunger category, while 8.1% were more likely to be in the moderate hunger category, and 8.7% were less likely to be in the little to no hunger category when compared to households under the statutory land tenure. A study in Bangladesh by Nasrin and Uddin [74] analyzed tenure systems that were classified as share tenants without land rights and cash tenants who held secure land rights. The study found higher food security in households that had secure land rights. Our results are in line with Ghebru and Holden [75], who demonstrated that tenure secure households, measured by the provision of land certificates, had a positive association with food security in Ethiopia. Furthermore, our findings complement those found by Mueller et al. [76] who studied the benefits of land reform programs for households, providing them with land titles to strengthen their land property rights on food security in Malawi. They demonstrated that food security of the households with more secure property rights improved in the long term. Apart from land property rights, an increase of land size in resettlement schemes also contributed to food security.

The results of our model showed that a one-hectare increase in land size is associated with being 2.4% less likely to be in the poor FCS status, 1.1% more likely to be in the borderline status, and 2.4% more likely to be in the acceptable FCS status. Similarly, the HHS model demonstrated that the probability of a household with one-hectare larger land size was reduced by 2% and 5% with regard to being in the severe hunger category and moderate hunger category, respectively while the probability of being in the little to no hunger category increased by 5.3%. One plausible explanation is that agriculture households with larger land size may have crop diversity, providing more nutritious crops when compared to households with smaller land size, who may highly consider cultivating only staple cereals. Githinji [77] studied how land influences household poverty levels in Kenya. The findings showed that an increase in land size reduced the probability of households being in the poor poverty levels. Furthermore, our finding is in agreement with that of Rammohan and Pritchard [78], who used ordered probit models to estimate if land holding was a determinant of household food and nutrition security in Myanmar. Their result indicated that an increase in land size enhanced household food security status. Similarly, our result is in alignment with that of Muraoka et al. [79], who analyzed the relationship between land access and food security in Kenya. They demonstrated that an increase in land size resulted in a rise in household food security.

### 5.4. Institutional Characteristics

The households that are members of a farming group or cooperative were indicated as being 8% less likely to be in the poor FCS category, while they had respectively 3.9% and 7.9% more chance of

being in the borderline and acceptable FCS categories than those who were not. The HHS revealed that membership to a farmer's organization reduced the probability of a household being in the severe hunger category by 6.9%, while such a household was 15.9% less likely to be in the moderate hunger and 17.5% more likely to be in the little to no hunger category. The results are in line with Nugusse et al. [80], who examined the association of cooperative and food security in northern Ethiopia. The study revealed that 21% of households with cooperative membership were food-insecure, while 35% of households without cooperative membership were food-insecure. Similarly, Wossen et al. [81] studied the effects of access to extension services and membership to cooperatives on household welfare in rural Nigeria. The results indicated that extension access and cooperative membership had a positive relationship with poverty reduction.

## 6. Conclusions

The main aim of this study was to examine the association of the chosen socioeconomic factors as influencers on food security. The study was conducted in 2016 in the southern province of Zambia. Food security was measured by the food consumption score (FCS) and the household hunger scale (HHS) indicators. From our sample, both the FCS and HHS ordered probit models' findings revealed that higher education levels of household head, increasing livestock incomes, secure land tenure, increasing land size, and group membership increased the probability of household food security.

Considering the fact that a larger number of households keep livestock based on cultural tradition, strengthening of livestock ownership and incomes should be prioritized. Support toward training and animal husbandly development with respect to environmental challenges and animal diseases may enhance the livestock production. According to the results of this study, we can expect that livestock development programs such as training of farmers in animal husbandry would improve livestock productivity and, thus, increase food security.

To improve household food and nutritional security in the long run, the development of food and land policies that are in accordance with the revealed determinants of food security may be recommended. The results of our study show that land tenure security increases food security. Thus, to increase food security, measures that would safeguard higher land security for households under customary tenure should be introduced. The most important in this respect is the implementation of a more effective land rights protection law. To speed up the process, stakeholders such as the national farmers union or local municipalities need to lobby the central government to implement a more effective law. Increased tenure security could be achieved, for example, through the inclusion of customary tenured households in land registration programs with legal recognition.

The size of land was found to have a positive relationship with food security. Therefore, pursuit of policies that help smallholder farmers with holdings of arable land, especially in customary land tenure, must be promoted. Recently, risks of some local traditional authorities not collaborating with communities within their authority in some instances gave rise to land grabbing. This is a case where the traditional leaders (chiefs) can decide to rent part of the land to an enterprise and make the land size of domestic farmers smaller. The decreasing farm size may affect the agricultural productivity of smallholder farms and limit their potential of attaining better food security.

Our findings demonstrate a positive impact of farming group membership on food security. Therefore, interventions to support organization and empowerment of existing informal and formal groups, especially through community mobilizing, should be encouraged by private and government organizations. Facilitation of official registration of farmers groups at agricultural district offices should be a priority. The registration must be planned beyond the current situation, where the majority of groups are only organized and oriented toward benefiting from programs such as the farmers input support. Only registered farming groups may provide training of members to help them improve the household food security status. Furthermore, farmers groups create opportunities for sharing of experiences among farmers and with other existing groups. Empowerment of farmers groups through adequate policy measures has the potential to improve household food security.

With regard to future research on farmers' education, from the methodological point of view, it would be interesting to include more variables representing knowledge acquisition other than the level of household head education in the survey and analysis, to help understand in more depth how receiving information helps decision-making toward food and nutritional security. Concerning land ownership studies, a focus on perceived tenure insecurity and inequalities among women and youth who are mostly reported as marginalized in traditional land with respect to land holding and agricultural output may be of interest for consideration. Regarding membership to farmer groups, an area of potential further research may focus on factors and barriers that motivate farmers to participate or not to participate in cooperatives. This study was limited by regional coverage within Zambia; however, the findings provide a fundamental base regarding the determinants of food security.

**Author Contributions:** W.N., methodology, investigation, writing—original draft preparation; M.B., methododology, data curation, formal analysis, writing—review and editing; J.B., conceptualization, methodology, writing—review and editing, supervision, project administration, funding acquisition.

**Funding:** This research was funded by Internal Grant Agency of Faculty of Tropical AgriSciences: grant number 20195006, and grant number 20195008.

**Conflicts of Interest:** The authors declare no conflict of interest.

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
