# Peer review of "Factors Associated with Household Food Security in Zambia"

_sustainability, doi:10.3390/su11092715_

Round 1

Reviewer 1 Report

The paper makes an empirical contribution to our understanding of food security and household attributes including education attainment, land tenure, and income, etc.  In the abstract, authors should highlight the novelty of the research.  It is well-known that food security is related to these household attributes; is the novelty that a study such as this has not been conducted with this particular study population?

Title should be a statement rather than a question.  Suggestion to include study population in title—authors should include key terms that will make the paper easier to identify the contents.

What is the central argument of the paper?  The value of research is that it supports or contradicts prior research.

Thorough edit needed.

Authors never establish the significance of investigated smallholder farmers’ food security. 

Introduction:

First sentence: report rates of food insecurity and undernourishment—absolute numbers are meaningless without knowing if rates are increasing or decreasing (i.e. the global population is also increasing).  Overall, the introduction is thin.  It should introduce what we generally know about determinants of food security (which are then detailed in the following literature review).  The last paragraph of the intro is awkward.  Typically, the structure of the paper is not described in this way.  The purpose of the introduction is to provide the readers the larger context, identify how it manifests in a particular way (which will be the subject of the study), and state the problem and why it is significant.  Include a statement of the argument.

Literature review

The review of the literature should begin with a summary / introductory paragraph.  Need to more clearly link the review of the literature with research variables in a comprehensive and critical way rather than simply a listing of various studies.  As is, it is written as a literature summary without a clear framing based on the context of this paper.  If smallholder farmers in Zambia is the study population, lit review should focus on (1) short, concise summary of lit re: food security and household characteristics; (2) the significance of smallholder farmers’ FS; (3) what we know and don’t know about Zambia smallholder farmers.  Conclude with a paragraph stating the research question, objectives, and study context.

Data and Methodology:

Is there any reason these four districts were chosen?

Reorganize study area section to group topics together—describe general population and livelihoods in one paragraph, land tenure policies in another, etc.

Data collection and sample: Who conducted the survey?  Need citation and more detailed summary re: food security in this area.  Why not stratified random sample?  IRB-approval?  What is the reliability/validity of the survey questions—was it based on a standardized questionnaire?  Are data secondary?  Or were the authors involved with primary data collection?  How were households within the villages sampled?  Could any household member participate or only the head of household?  Was a translator involved?  How were data recorded?  Need a more detailed summary of variables. 

First paragraph of section 3.3.1 should move to literature review.  The description of FCS and HHS should be moved under a section called ‘research variables’ or ‘survey instrument’ because it is methods not analysis.

There is no need for a statistical tutorial re: ordered probit models.  Rather, the authors should simply describe the dependent and independent variables and how they were modeled.   Dependent and explanatory variables should be individually listed and it should be clarified how the variables were grouped.   Need to describe the models—full model? Partial model? Bivariate?  Control variables? 

Description of model variables should be moved to results.

Results should begin with a summary paragraph describing the study sample in a logical order.  How did the sample differ across the 4 areas?  Rather than simply converting the table into text format, the authors should synthesize findings into a more narrative summary (while being careful to reserve interpretations for the discussion section).

Table 2 and 3 need to be described in paragraph format.  Pull out key findings.  Summarize.  Use period (.) not comma (,) to demarcate decimal places.  Indicate p-value.  Be consistent with the number of decimal places reported.  Do these tables report the full model (all variables included) or partial?

Use past tense.

Discussion

Need to report p-value rather than se—are all these findings significant?  They aren’t and the authors should report what is statistically significant and also discuss which variables are not.

If all of these variables have already been studied—what is the unique and valuable about the contribution of this paper?  What are the implications of this study?  Need to question findings: what are the limitations, biases, etc of this study? 

Conclusion should draw back to the contextual significance of the study.  What is the more general takeaway?  What is the larger contribution to the state of the research?

Author Response

Dear Editor,

We are grateful for the feedback on our manuscript as it contributed to improvement of the paper. We have considered and addressed all the reviewer comments. The responses and adjustments to the comments are highlighted in red.

Reviewer 1

Comments and Suggestions for Authors

The paper makes an empirical contribution to our understanding of food security and household attributes including education attainment, land tenure, and income, etc.  In the abstract, authors should highlight the novelty of the research.  It is well-known that food security is related to these household attributes; is the novelty that a study such as this has not been conducted with this particular study population?

The novelty of this study is that little or limited studies have focused on the influencers of food security in Zambia, therefore, our study fills this gap in knowledge.

Title should be a statement rather than a question.  Suggestion to include study population in title—authors should include key terms that will make the paper easier to identify the contents.

The authors have revised the title.

What is the central argument of the paper?  The value of research is that it supports or contradicts prior research.

This study was carried out to identify factors associated with increased Food insecurity in Zambia. Second, to the best of the researchers’ knowledge, there is scarcity in the literature on this topic, yet it is a critical subject that policymakers need to understand in Zambia. Therefore, this study aims to consolidate past studies to add incremental contribution to Zambia and other nations faced with food insecurity challenges.

Thorough edit needed.

The authors have made edits were appropriate.

Authors never establish the significance of investigated smallholder farmers’ food security. 

In Zambia, agriculture employs 72% of the country’s labour force, with more than 60% residing in rural areas . The predominant livelihood activity is smallholder farming mainly cultivating maize and livestock raring. The smallholder farmers are also faced with higher levels of food insecurity ( Highligted under the study area). The authors have added these aspects under the introduction section.

Introduction:

First sentence: report rates of food insecurity and undernourishment—absolute numbers are meaningless without knowing if rates are increasing or decreasing (i.e. the global population is also increasing).  Overall, the introduction is thin.  It should introduce what we generally know about determinants of food security (which are then detailed in the following literature review).  The last paragraph of the intro is awkward.  Typically, the structure of the paper is not described in this way.  The purpose of the introduction is to provide the readers the larger context, identify how it manifests in a particular way (which will be the subject of the study), and state the problem and why it is significant.  Include a statement of the argument.

 Food insecurity and undernourishment are on the rise worldwide, from an estimated 777 million people in 2015 to 815 million people in 2016 . This increase is a global concern in achieving the second sustainable development goal, which calls for a commitment to end hunger, reduce food insecurity and improve nutrition by 2030.

The authors have added general description of  determinants of food security in the introduction section and furthermore, discussed them in detail under the conceptual framework.

The significance and argument statement of the study has also been highlighted in introduction section as suggested.

Literature review

The review of the literature should begin with a summary / introductory paragraph.  Need to more clearly link the review of the literature with research variables in a comprehensive and critical way rather than simply a listing of various studies.  As is, it is written as a literature summary without a clear framing based on the context of this paper.  If smallholder farmers in Zambia is the study population, lit review should focus on (1) short, concise summary of lit re: food security and household characteristics; (2) the significance of smallholder farmers’ FS; (3) what we know and don’t know about Zambia smallholder farmers.  Conclude with a paragraph stating the research question, objectives, and study context.

 The determinants that are found to have been studied in previous related studies are considered in the conceptual framework.

The significance of smallholder farmers’ FS has been included in the Introduction section and emphasized in the study area.

The objective of the study and study content have been considered in the concluding part of the introduction.

Data and Methodology:

Is there any reason these four districts were chosen?

 The province and districts were purposively selected. The region was selected because even though the area is regarded as the food basket of the country, the population still faces food insecurity, which makes it suitable for the study (described under data collection and sample)

Reorganize study area section to group topics together—describe general population and livelihoods in one paragraph, land tenure policies in another, etc.

 The authors have reorganized the study area sections as recommended

Data collection and sample: Who conducted the survey?  Need citation and more detailed summary re: food security in this area.  Why not stratified random sample?  IRB-approval?  What is the reliability/validity of the survey questions—was it based on a standardized questionnaire?  Are data secondary?  Or were the authors involved with primary data collection?  How were households within the villages sampled?  Could any household member participate or only the head of household?  Was a translator involved?  How were data recorded?  Need a more detailed summary of variables. 

Primary data collection was conducted by authors and local trained enumerators. The local enumerators did the translation were it was required.

Research consent was obtained from the district commissioners office which is adequate for this study.

A structured questionnaire was used for primary data collection.

Three villages were selected per tenure system in each district, using a systematic approach that was guided by the following key features: i) villages in different locations and ii) villages with comparable tenure systems and the households simple randomly sampled.

The data was recorded on the paper questionnaires (pen-and-paper personal interview) and later coded on microsoft excel spreadsheets.

First paragraph of section 3.3.1 should move to literature review.  The description of FCS and HHS should be moved under a section called ‘research variables’ or ‘survey instrument’ because it is methods not analysis.

 The first paragraph of section 3.3.1 has been moved to literature review as recommended

The description of FCS and HHS has been moved under a section called ‘research variables’ as recommended

There is no need for a statistical tutorial re: ordered probit models.  Rather, the authors should simply describe the dependent and independent variables and how they were modeled.   Dependent and explanatory variables should be individually listed and it should be clarified how the variables were grouped.   Need to describe the models—full model? Partial model? Bivariate?  Control variables? 

 The model, the dependent and independent variables are described under section 3.3.2, specifically listing the classification of variables into groups.

Description of model variables should be moved to results.

 Description of model variables have been moved to results as recommended

Results should begin with a summary paragraph describing the study sample in a logical order.  How did the sample differ across the 4 areas?  Rather than simply converting the table into text format, the authors should synthesize findings into a more narrative summary (while being careful to reserve interpretations for the discussion section).

A combined sample describing the four study areas has been put at the beginning of the results section as recommended.

Table 2 and 3 need to be described in paragraph format.  Pull out key findings.  Summarize.  Use period (.) not comma (,) to demarcate decimal places.  Indicate p-value.  Be consistent with the number of decimal places reported.  Do these tables report the full model (all variables included) or partial?

Table 2 and 3 are described in paragraph format under the section Discussion. The commas have been changed to periods and number of decimal places revised and consistent. The tables report the full model (all variables included).

Use past tense.

The authors have adjusted were appropriate

Discussion

Need to report p-value rather than se—are all these findings significant?  They aren’t and the authors should report what is statistically significant and also discuss which variables are not.

 Note: *** p<0.01, ** p<0.05, * p<0.1. The average marginal effects are reported with the standard errors in parentheses (placed under Table 2 and 3). The authors considered the P- values for interpretation and not the SE.  Not all variables are statistically significant. We have considered those statistically significant and some variables that are unexpectedly not statistically significant.

If all of these variables have already been studied—what is the unique and valuable about the contribution of this paper?  What are the implications of this study?  Need to question findings: what are the limitations, biases, etc of this study? 

The uniqueness and value of the paper especially for Zambia is that this paper contributes as a good base for in depth studies in this direction. The authors have suggested some direction for future studies in the conclusion section regarding the different determinants.

The study was limited by regional coverage within Zambia; however, the findings provide a fundamental base regarding the determinants of food security. 

Conclusion should draw back to the contextual significance of the study.  What is the more general takeaway?  What is the larger contribution to the state of the research?

The authors have highlighted the different determinants of the food security and the focus that policy makers in Zambia can take as recommendations.

Yours Faithfully,

Authors.

Reviewer 2 Report

This is a very well written manuscript.  There are some editorial issues here and there.  The approach and analysis are sound.

My only significant issue is language that suggests causality.  The title needs a tweak, and suggests causation, which can't be shown with a cross-sectional study.  Similarly, some of the language in the conclusion also suggests causation (lines 429, 434, etc.).  I would recommend softening the language to "associated with," etc.

An issue for you to consider is the extensive literature review, and then the revisiting of this in the results section. The results could be considerably shortened, and the discussion of the literature could moved back to the conclusions.

Interesting paper!

Author Response

Dear Editor,

We are grateful for the feedback on our manuscript as it contributed to improvement of the paper. We have considered and addressed all the reviewer comments. The responses and adjustments to the comments are highlighted in red.

Reviewer 2

Comments and Suggestions for Authors

This is a very well written manuscript.  There are some editorial issues here and there.  The approach and analysis are sound.

My only significant issue is language that suggests causality.  The title needs a tweak, and suggests causation, which can't be shown with a cross-sectional study.  Similarly, some of the language in the conclusion also suggests causation (lines 429, 434, etc.).  I would recommend softening the language to "associated with," etc.

The authors have revised the title of the paper to avoid causation. We have further softened the language on recommended parts.

An issue for you to consider is the extensive literature review, and then the revisiting of this in the results section. The results could be considerably shortened, and the discussion of the literature could moved back to the conclusions.

The authors have deleted some of the literature review that was considered in conceptual framework and revisited in results section.

Interesting paper!

Yours Faithfully,

Authors.